# Utilizing Proteomic Approach to Analyze Potential Antioxidant Proteins in Plant against Irradiation

**DOI:** 10.3390/antiox11122498

**Published:** 2022-12-19

**Authors:** Ming-Hui Yang, Yi-Shan Lu, Tzu-Chuan Ho, Daniel Hueng-Yuan Shen, Ying-Fong Huang, Kuo-Pin Chuang, Cheng-Hui Yuan, Yu-Chang Tyan

**Affiliations:** 1Department of Medical Education and Research, Kaohsiung Veterans General Hospital, Kaohsiung 813, Taiwan; 2Center of General Education, Shu-Zen Junior College of Medicine and Management, Kaohsiung 821, Taiwan; 3Office of Safety, Health and Environment, Kaohsiung Medical University, Kaohsiung 807, Taiwan; 4Department of Medical Imaging and Radiological Sciences, Kaohsiung Medical University, Kaohsiung 807, Taiwan; 5Department of Nuclear Medicine, Kaohsiung Veterans General Hospital, Kaohsiung 813, Taiwan; 6Department of Nuclear Medicine, Kaohsiung Medical University Hospital, Kaohsiung 807, Taiwan; 7Graduate Institute of Animal Vaccine Technology, College of Veterinary Medicine, National Pingtung University of Science and Technology, Pingtung 912, Taiwan; 8Mass Spectrometry Laboratory, Department of Chemistry, National University of Singapore, Singapore 119077, Singapore; 9School of Medicine, Kaohsiung Medical University, Kaohsiung 807, Taiwan; 10Graduate Institute of Medicine, College of Medicine, Kaohsiung Medical University, Kaohsiung 807, Taiwan; 11Department of Medical Research, Kaohsiung Medical University Hospital, Kaohsiung 807, Taiwan; 12Center for Cancer Research, Kaohsiung Medical University, Kaohsiung 807, Taiwan; 13Research Center for Precision Environmental Medicine, Kaohsiung Medical University, Kaohsiung 807, Taiwan

**Keywords:** plants against stress, anti-oxidant, gamma-ray irradiation, flavonoid, reactive oxygen species, plant defense response

## Abstract

Gamma-ray irradiation is an effective and clean method of sterilization by inactivating microorganisms. It can also be applied to induce anti-oxidants for future application. In this study, the mung bean (*Vigna radiata*) was exposed to gamma-ray irradiation under the dose of 0, 5 or 10 kGy. With increasing irradiation doses, the concentrations of malondiadehyde decreased while the levels of total flavonoids and DPPH (1,1-diphenyl-2-picrylhydrazyl) radical scavenging activity increased. It has been shown that consuming flavonoids can provide protective effects. In addition, proteomic analysis identified several proteins having anti-oxidant activities in the 5 kGy irradiated group. These proteins are Apocytochrome *f*, Systemin receptor SR 160, DELLA protein DWARF8, DEAD-box ATP-dependent RNA helicase 9, ζ-carotene desaturase (ZDS), and Floral homeotic protein AGAMOUS. Our findings indicate that plants contain a variety of phytochemicals and antioxidant proteins which may effectively prevent oxidative stress caused by irradiated peroxidation.

## 1. Introduction

‘Ionizing radiation’, abbreviated as ‘radiation’ or ‘irradiation’, describes a capacity to ionize (to remove electrons from) atoms or molecules and is categorized by the nature of the particles (i.e., α particles, β particles, neutron, high speed electron, high speed proton and other particles) or electromagnetic waves (i.e., X-ray and γ-ray) that create the ionizing effect. Based on different ionization mechanisms, radiation is also grouped as direct or indirect radiation. Indirect radiation includes X-rays and γ-rays, which are produced by extranuclear and intranuclear processes, respectively. They may cause chemical and/or biological damage through the interaction of charged particles within the medium being traversed. The use of ionization radiation has to consider the nuclear safety including not having chemical reactions nor nuclear dissociations. As a result, only elections with energy less than 10 MeV or X-ray and γ-rays with energy less than 5 MeV can be chosen. Gamma ray, an ionizing radiation, consists of high-energy photons and is biologically hazardous. It mainly decays from radionuclides such as Co-60 or Cs-137. Today, Co-60 is most frequently utilized in food irradiation for sterilization due to its gamma-ray energy levels equal to 1.17 and 1.33 MeV, which are under 5 MeV and compliant with regulations for the use on sterilization [1]. The gamma ray has a high penetration ability, thus the gamma ray does not induce radioactivity in food or over the packaging. Only very little dosage of gamma ray might be absorbed and then transferred as heat.

Currently, high-energy gamma-ray irradiation is used to prevent fungal and bacterial infection of seeds and foodstuffs to reduce economic losses during transportation [2]. During the process of plant seed sterilization, gamma-irradiation can induce reactive oxygen species (ROS) generation in plants. These ROS can be in the forms of superoxide ion (O2·^−^), peroxyl (ROO·), alkoxyl (RO·), hydroxyl (OH·), nitric oxide (NO·), or hydrogen peroxide (H_2_O_2_) [3]. ROS may provoke several harmful effects, such as damages on nucleic acids (DNA or RNA), cell membranes, proteins, and chlorophylls [4]. High doses of irradiation on plants can cause oxidation stress [5]. The oxidation stress may influence plant growth, in addition to causing leaf chlorosis, necrosis or apoptosis [6,7]. Even more serious to make the apoptotic-like effects such as nuclear fragmentation and DNA laddering [8]. Furthermore, biomacromolecules are mainly sustained in aqueous environments. When radiation energy works on water molecules, water itself can be dissociated or excited. This will generate harmful free radicals causing damages on biomacromoelcules within the aqua environment. Radiation can affect the stability of DNA or RNA. DNA contains genetic information using genetic codes and can be damaged by the radiation leading to DNA breaks or error-prone repairs. When these happen, cell mutation or death may occur [9]. The process of sterilization starts from the molecules being irradiated to produce ionization and free radicals. After a series of chemical reactions, the biochemical changes cause damages on the physiology. Next, the metabolism slows down or even being arrested causing the death of the microorganisms. As a result, the sterilization is completed. Moreover, gamma-ray irradiation has been chosen for food sterilization due to the following reasons: improved hygienic quality, reduced the loss from microbial contamination, and decreased the use of chemical fumigants and preservatives. According to the literature, radiation decontamination used energy ranging from 3 to 10 kGy can be a viable alternative to chemical fumigants [10]. The opportunities of recontamination are also reduced because gamma-ray irradiation can be performed after packing. However, plants may maintain or enhance the antioxidant properties in order to counteract damage from gamma-ray irradiation. In the gamma-ray irradiation process, the phenolic content and the free radical-scavenging activity are increased in plants, which can indirectly protect plants against radiation [11]. Plants can endure the harsh environmental conditions through scavenging toxic oxygen species. This enables the resistance to environmental stresses. In order to survive and prevent irradiation damage, plants have developed several anti-oxidant mechanisms, such as transferring ROS into less-toxic products [12]. At this moment, highly expressed anti-oxidants can increase the endurance of plants against environmental threat [13]. From the literature, the changes of anti-oxidation enzyme expressed levels are related with radiation [14]. It has been confirmed that anti-oxidants in edible plants, such as beans, can be induced under stressful environment [15]. However, the effects of digesting these edible plants are unrevealed.

Proteomics is to characterize the entire proteins produced within a cell, tissue, or organism at a specific time [16]. Moreover, it is considered as an alternative means to elucidate the message from genomic sequences in aspects of the structure, function, and molecular pathways. Thus, proteomics becomes a tool utilized to understand biological processes through a comprehensive and systematic analysis of proteins expressed in a cell or tissue [17]. Mass spectrometry (MS) is one of the most essential tools in proteomics. The protein structure details, including peptide sequences and molecular weights can be acquired by MS. Such information is utilized to identify proteins using nucleotide and protein databases. In addition, post-translational modifications of proteins can be identified.

Plants survive from environmental stresses mainly due to their ability to recognize and adapt in addition to regulating the expression of stress-responsive genes/proteins. In this study, an important economic and nutritional plant, mung bean (*Vigna radiata*), was chosen to be the experimental model. Mung beans are fast-growing and mainly cultivated in tropic and subtropical areas. Mung beans are high in carbohydrate, minerals and vitamins while low in fat. It can be eaten cooked or sprouted. The nutrients in the mung beans are enhanced when it is sprouted. Mung bean sprout is abundant in vitamin C, flavonoids and phenolics, and has high antioxidant activity [18]. The use of gamma radiation on grain has been debatable. Some argue that this type of ionizing radiation can induce mutations being effectively beneficial in plant breeding [19]. Some suggests gamma radiation caused high mortality and being harmful for germination [20]. In this study, the anti-oxidant effect of gamma radiation was evaluated and a proteomic analysis of the mung bean after gamma-ray irradiation was also explored.

## 2. Materials and Methods

### 2.1. Plant Growth Conditions and Gamma-Ray Irradiation

Mung beans were obtained from Putzu Farmers’ Association (Chiayi, Taiwan). Individual mung beans were prepared by selection of equal size with an intact unblemished seed coat. Gamma-ray irradiation was performed by exposing the seeds to an irradiation dose of 5 or 10 kGy in a gamma ray cell with ^60^Co source (dose rate: 4 kGy/h) installed at the Institute of Nuclear Energy Research, Longtan, Taiwan. The seeds were irradiated for 1.25 and 2.5 h to obtain irradiation dose of 5 and 10 kGy, respectively. Dose rate was determined by using an alanine and electron paramagnetic resonance (EPR) dosimetry system. The seeds without gamma-ray irradiation were used as controls. The irradiated seeds along with the parental control were soaked in distilled-deionized water (D.I. water) at room temperature for 12 h to make the outer covering chap. Next, the seeds were incubated in 15 cm dishes with random-sampling, and grown at room temperature with normal photoperiods (25 °C, 12 h of sunlight, and 12 h of uninterrupted darkness). To provide the whole nutrition to the seeds, D.I. water was supplied every day. All seeds of each treatment were observed for seven days, collected and frozen using liquid nitrogen, and then stored at −80 °C for the following biochemical studies.

### 2.2. Protein Extraction

Each frozen mung bean sample (0.1 g) was ground into fine powders in liquid nitrogen using a pestle and a mortar. Two milliliter ice-cold extraction buffer (CelLytic^TM^ P, Sigma St. Louis, MO, USA) was added into the mortar and ground thoroughly with the pestle. The protease inhibitor (Protease Inhibitor Cocktail, Sigma) was used to improve the yield of intact proteins. The mixture was centrifuged at 12,000× *g* for 10 min at low temperature. Next, the supernatant was collected and passed through a 0.8 nm mesh filtration membrane. The resulting solution was collected and stored at −80 °C until further analysis.

### 2.3. Protein Concentration Measurement

The protein quantitation was carried out using a commercial supply (Quant-iT™ Protein Assay Kit, Thermo Scientific, Waltham, MA, USA). The Quant-iT protein reagent was diluted with a ratio of 1:200 with Quant-iT protein buffer as a working solution. The protein standard solution was prepared by adding 10 μL of BSA standard to 190 μL of working solution to form a final analytic volume of 200 μL. Similarly, the sample solutions were prepared using 180–199 μL of working solution with 1–20 μL of unknown samples to form a final analytic volume of 200 μL. The standard solutions and sample solutions were loaded into the Qubit^®^ assay tubes (Thermo Scientific, Waltham, MA, USA), the assay was performed at room temperature for 15 min, and then the fluorescence reading was measured using a Qubit^®^ fluorometer. Duplicates of the standards and the unknown samples were performed.

### 2.4. Determination of Malondialdehyde Concentration

Lipid peroxidation was calculated by detecting the amount of malondialdehyde (MDA) with Cayman’s Kit (Thiobarbituric Acid Reactive Substances Assay Kit, Cayman Chemical, Ann Arbor, MI, USA). Thiobarbituric acid (TBA, 530 mg) was accurately weighed and dissolved in 50 mL diluted acetic acid solution. Next, diluted sodium hydroxide was added with stirring. Unknown samples or standards were mixed with TBA SDS solution in 1:1 ratio to make 200 microLiter solution per sample before adding 4 mL of TBA color reagent. Such mixtures were then incubated in boiling water for an hour to react. After cooling, the samples were centrifuged at 1600× *g* at 4 °C for 10 min, and placed at room temperature for 0.5 h. The measurement of absorbance at 520 nm was recorded for each sample using an ELISA reader (MULTISKAN EX, Thermo Scientific, Waltham, MA USA).

### 2.5. Determination of Flavonoids Content

The total flavonoid content was measured spectrophotometrically by forming a complex with AlCl_3_ (aluminum chloride, anhydrous, granular 99%, Alfa Aesar, Ward Hill, MA, USA). In brief, 100 μL of each group of the prepared sample solution was incubated with 100 μL of 2% (*w*/*v*) AlCl_3_ methanolic solution for 10 min. Next, the absorbance at 450 nm of each resulting solution was measured using an ELISA microplate reader (MULTISKAN EX, Thermo). A series of rutin (rutin trihydrate, Sigma, St. Louis, MO, USA)) solution was prepared in triplicate and diluted to concentrations of from 6.25 to 250 μg/mL and used as standards.

### 2.6. Free Radical Scavenging Activity Using DPPH

The free radical scavenging assay was carried out according to the method illustrated by Khattak and coworkers with slight modification [11]. The antioxidant activities of the ethanolic extract samples were determined using 1,1-diphenyl-2-picrylhydrazyl (DPPH) radical method. DPPH is a stable purple free radical with an absorption at 520 nm. The free radical-scavenging assay is measuring the de-coloration when reduction occurred. In short, each extract (50 μL) was mixed into 750 μL of DPPH ethanolic solution (freshly prepared) before diluted with an extra of 200 μL ethanol. A control was prepared by mixing 750 μL of DPPH ethanolic solution and 250 μL of ethanol. The absorbance at 520 nm of each sample was measured using an ELISA microplate reader (MULTISKAN EX, Thermo). Quintuplicates of the control and the unknown samples were performed.

The DPPH radical scavenging activity (%) was calculated using the following formula:

$$\text{Radical scavenging activity (\%)}=(1-\frac{A_{i}}{A_{o}})\times 100$$
where *A_o_* and *A_i_* were the absorbances of control and test samples, respectively.

### 2.7. Protein Identification by Mass Spectrometry

In protein identification, proteins were identified by reversed phase-nano-ultra performance liquid chromatography-electrospray ionization with tandem mass spectrometry (RP-nano-UPLC-ESI-MS/MS). A tryptic digestion was prepared using Mass Spectrometry Grade trypsin (Trypsin Gold, Promega, Madison, WI, USA). Each prepared sample of 0.1 mL was added to a plastic Eppendorf tube before being reduced with 1 M dithiothreitol (DTT, USB Corporation) of 10 μL in 25 mM ammonium bicarbonate at 37 °C for three hours. Next, the mixture was further alkylated with 1 M iodoacetamide (IAA, Amersham Biosciences, Chicago, IL, USA) of 25 μL in 25 mM ammonium bicarbonate in the water bath at 25 °C in the dark for another 30 min. Subsequently, trypsin of 0.1 μg/μL was added and incubated in the water bath at 37 °C for at least 12 h before quenching by adding formic acid of 2 μL.

Mass spectroscopic peptide separation and sequencing were performed on a nano-UPLC System (nanoACQUITY UPLC, Waters, Milford, MA, USA) coupled to an Ion Trap Mass Spectrometer (LTQ Orbitrap Discovery Hybrid FTMS, Thermo, San Jose, CA, USA). Each digested sample of 2 μL was separated using a reverse-phase Symmetry C18 column (5 μm, 180 μm × 20 mm) at 400 nL/min flow rate. Peptides were eluted with a linear gradient of 99% Buffer A (100% double-distilled water with 0.1% formic acid) to 85% Buffer B (100% acetonitrile with 0.1% formic acid) with a total cycle time of 100 min. Subsequently, the peptide was separated through a flow splitter onto a C18 Microcapillary Column (BEH C18, 1.7 μm, 75 μm × 100 mm) for direct infusion at 400 nL/min through a nano-spray tip into the ESI-MS/MS with a distal 2.1 kV spraying voltage with heated capillary (200 °C). The data were acquired in the full-scan mass spectra of *m*/*z* 400–2000 and followed by four data-dependent tandem mass spectra with collision energy settings of 35%.

### 2.8. Quality Control of Mass Spectrum

Tandem mass spectrometry data were analyzed by the database search software SEQUEST (Bioworks 3.1, ThermoFinnigan, San Jose, CA, USA) to interpret MS/MS spectra. DTA files were generated from product ion scan data (threshold intensity set at 10,000). The quality of each proteomic mass spectrum was confirmed by the cross-correlation (XCorr) score. The XCorr score is a correlation means for the measured and theoretical MS/MS spectra. Nevertheless, XCorr values were typically higher for well-matched peptideswith large molecular weights, and lower for smaller peptides. In this study, XCorr scores of at least 1.9, 2.5, and 3.5 for the 1+, 2+, and 3+ ions were regarded as high confidence of protein identification. The criterion for manual validation required an easily observable series of least 4 y ions, and then such protein was assessed as present in the sample.

### 2.9. Proteomic Identification by Database Searching

In database search, the obtained peptide mass fingerprints were submitted to MS/MS ion search in Mascot server (Ver. 2.3, Matrix Science, Boston, MA, USA) using UniProtKB/Swiss-Prot databases (downloaded from NCBI website) [21]. In order to denote a protein as an unambiguous identification, the following criteria were used: cleavage by trypsin enzyme, two missed cleavage allowed; monoisotopic peptide masses with ±0.5 Da peptide mass tolerance and 0.5 Da fragment mass tolerance; peptide charge of 1+, 2+ and 3+; and variable modifications allowed of Carboxymethyl (C), Deamidated (NQ), Oxidation (M) and Phospho (STY). Each spectrum of fragment ion was checked against the same database and the identification was confirmed if such correspondence was found. As the MOWSE score was greater than 30, the protein identification was defined as positive and considered as significant (*p* < 0.05).

### 2.10. Statistical Analysis

All data used the SigmaStat statistical software (Jandel Science Corp., San Rafael, CA, USA). The data are presented as mean ± standard deviation. The statistical significance was present as *p* < 0.05.

## 3. Results and Discussions

### 3.1. Plant Appearance and Growth Condition

The appearance and growth condition of the gamma ray irradiated seeds along with the parental control were shown in Figure 1 and Figure 2. It was displayed significant differences in height and color after gamma-ray irradiation with various doses for four days. The control group grew fast with apparently individual diversity and the seed coat appeared in green. The mean height of the plants was 1.80 ± 0.66 cm in Day-1, 5.10 ± 1.22 cm in Day-3 and 5.33 ± 1.14 cm in Day-4. In contrast, the seeds irradiated with 5 kGy grew slower and the seed coat appeared in yellowish-brown. The mean height of the plants was 0.83 ± 0.13 cm in Day-1, 1.13 ± 0.24 cm in Day-3 and 1.19 ± 0.21 cm in Day-4. The seeds irradiated with 10 kGy did not even germinate after gamma-ray irradiation for 4 days and the seed coat appeared as yellowish-brown. The results are agreement with the findings of Mandar Sengupta et al. They reported that damage caused by gamma-ray irradiation resulted in decrease of final germination percentage and seedling height [22].

### 3.2. Lipid Peroxidation, Flavonoid Content and Free Radical Scavenging Activity

Ionizing radiation can damage organisms by direct or indirect mechanisms. When ionizing radiation is absorbed in biological compositions, it can work on vital targets in cells. Alternatively, it might interact with molecules (especially water) within cells, to produce free radicals capable of diffusing to far distance and damaging different components [23]. Free radicals are specified as molecules with an unpaired electron in the outer orbit, which are almost always unstable and very reactive. Some ROS are well-known oxygen-centered free radicals, such as O_2_·^-^, ROO·, RO·, OH· and NO·. In addition, free radicals of hydroxyl and the alkoxyl are extremely reactive and can rapidly attack the molecules in neighboring cells, which most likely results in unavoidable damage. In contrast, the superoxide anions, lipid hydroperoxides and nitric oxide are less reactive [24].

Flavonoids are reported as having various biological activities, including anti-oxidative and free radical scavenging activities. Previous literature has shown that flavonoids can inhibit the formation of MDA due to their scavenging action towards hydroxyl radical precursors, i.e., superoxide anions [25]. Furthermore, flavonoids have been demonstrated to restrain lipid oxidation via scavenging superoxide anions which set off the free radical chain reactions generating lipid oxidation.

In our studies, the results of the lipid peroxidation product of MDA, and total flavonoid content after gamma-ray irradiation on mung beans with various irradiation doses are shown in Figure 3 and Figure 4, respectively. In Figure 3, the MDA concentrations were 37.54 ± 3.41, 10.17 ± 2.17 and 6.41 ± 1.39 μmole after the irradiation with doses of 0, 5 and 10 kGy, respectively. In Figure 4, the concentration changes of flavonoids were 14.32 ± 1.44, 87.65 ± 5.77 and 106.98 ± 4.09 μg/mL after irradiation with a dose of 0, 5 and 10 kGy, respectively. A positive correlation was found with increased total flavonoid content after higher irradiation doses. In particular, it significantly increased after the irradiation of 10 kGy. Nevertheless, the concentration of the lipid peroxidation product of MDA decreased when the total flavonoid content increased. Flavonoids are considered as scavengers. Their antioxidant properties are acquired via the scavenging of superoxide anions. Meanwhile, preventing lipid oxidation by scavenging superoxide anions [25]. The result also showed that the control group had the highest level of lipid oxidation. The most plausible explanation for the results might be that the plants were adversely affected after gamma-ray irradiation. According to the height of plants, there was less cell proliferated, and less membrane lipid generated. Subsequently, the concentration of lipid peroxidation product of MDA was decreased.

The results of the DPPH radical scavenging activity after gamma-ray irradiation on mung beans with various irradiation doses are shown in Figure 5. They were 39.52 ± 4.95%, 52.77 ± 0.93% and 58.44 ± 3.74 after radiation doses of 0, 5 and 10 kGy, respectively. According to the previous literature [26], the free radical scavenging activity raised when the flavonoid content increased. In our studies, the DPPH radical scavenging activity increased with an increasing irradiation dose. The presumable reason might be that flavonoids can reduce free radical formation and scavenge free radicals [24]. Therefore, the antioxidant activity will improve when the flavonoid content increases. Flavonoids are polyphenolic compounds, which consists of 15 carbon atoms organized within a diphenylpropane (C_6_-C_3_-C_6_) skeleton. Flavonoids are products resulted from the phenylpropanoid metabolic pathway in which phenylalanine, the amino acid, is used to produce 4-coumaroyl Co-A, and subsequently are combined with malonyl-Co A to yield the backbone of flavonoids. Gamma-ray irradiation causes the increase of free radicals in plants, but flavonoids have biological abilities to scavenge superoxide, hydroxyl and lipid hydroperoxides [27]. In this manner, flavonoids are considered as scavengers because they are capable of hydrogen-donating, which can prevent free radical generation and even stabilize the free radicals. Increasing the flavonoid content in plants can scavenge free radicals and reduce lipid oxidation.

### 3.3. Functional Classification and Subcellular Localization Prediction of Gamma Ray Responsive Proteins

In order to investigate the effect of gamma-ray irradiation on mung beans, a proteomic approach was performed using RP-nano-UPLC-ESI-MS/MS analysis and database searching. In the past, the aim of proteomic analysis was to analyze a large number of proteins in a short period of time. However, in this experiment, we are working on finding the key differential expressed proteins. Thus, a higher threshold was set for this purpose. In this study, we adapted two mass spectrometry analysis software: Bioworks and Mascot. The Bioworks workstation was used to assess the quality of the mass spectra in this experiment. The fragmentation spectra obtained were analyzed using Mascot software after being examined as high quality mass spectra. Next, protein functions were annotated using UniProtKB/Swiss-Prot databases. As a result, there are not a lot of proteins were identified in this experiment, but all of them were highly reliable.

At the initial proteomic mass spectrometry analysis of plant proteins at 0 and 5 kGy irradiation, most of them were identified at the minimal confidence level. Only a small portion of searches yielded significant matches based on the inclusion criteria. After data comparison using Mascot analysis software, a total of 181 proteins were identified in the 0 kGy group, 6 of which were highly specific proteins, and a total of 228 proteins were identified in the 5 kGy irradiation group, 12 of which were high-specificity proteins. Among them, six specific proteins exhibited significant differences in the treatment group.

As shown in Table 1, the six up-regulated proteins identified with higher confidence levels (at least four unique peptide sequences matched). These six up-regulated proteins are Apocytochrome *f,* Systemin receptor SR 160, DELLA protein DWARF8, DEAD-box ATP-dependent RNA helicase 9, ζ-carotene desaturase (ZDS), and Floral homeotic protein AGAMOUS. Since there is no antibody for plant protein for verification, only differential analysis can be carried out in this proteomic analysis. In addition, the relevant protein information and function can still be analyzed by the plant protein database. The functions of these six proteins are mostly increasing grain yield at the expense of straw biomass, and are increasing stress resistance in plant growth.

Apocytochrome *f* constitutes cytochrome *b_6_f* complex, which mediates the transfer of electrons between the two photosynthetic reaction center complexes, from Photosystem II (PSII) to Photosystem I (PSI). The plants exposed to ionizing radiation might receive direct and indirect (oxidative stress) deleterious effects. Such effect depends on the radiation dose applied, which leads to differential gene expression. In Gicquel’s study, plants with low dose of irradiation (without visible developmental phenotype changes) showed increased expression of cytochrome *b_6_f* complex [28]. In Suorsa’s study, cytochrome *b_6_f* complex is involved in protecting plants from the low light with short peaks of high light. This can prevent the electron transfer chain from hyper-reduction and, as a result, protect PSI from photodamage [29]. When plants were exposed to high light (the photon flux density was comparable to full sunlight), the sizes of the high light-grown plants were only one-third of those grown in low light (about 1/10 of the photon intensity). Furthermore, the protein expression of cytochrome *f* increased in the high light group [30]. Since the cytochrome *f* protein expression level generally correlates with the sum of the cytochrome *b_6_f* complex, and the cytochrome *b_6_f* complex is involved in light protecting, the increased expression of Apocytochrome *f* may provide a protection effect.

Systemin receptor 160 (SR160) belongs to a leucine-rich repeat receptor kinase (LRR-RK) family and is also a homologue of Brassinosteroid Insensitive 1 (BRI1, a brassinosteroid receptor) since systemin and brassinosteroid are probably perceived by different regions of the same receptor [31]. Nevertheless, this (one receptor with two distinctive functions and names) is still in debate. Malinowski and coworkers reported that SR160 is just a protein having remarkable similarity to BRI1 and has a similar molecular weight, glycosylation pattern, and antigenic epitope [32]. They demonstrated that BRI1 can bind to systemin, but such binding does not transduce the signal into the cell. The perception of systemin through the receptor SYR1 (one of the LRR-RKs) is important for defense against insect herbivory [33]. BRI1 recognizes brassinosteroids, which are essential phytohormones in regulating plant growth and development [34]. Systemin is involved in responding to insect attacks and play an important role in mechanical wounding. Upon the perception of systemin by SR160 (stress perception), several characters including fluxes of ions, kinases (Mitogen-activated protein kinases 1–3 and calcium-dependent protein kinases), organic compound (ethylene), ROS, and jasmonic acid (JA) are involved [35]. Plants overexpressing the prosystemin, a precursor of systemin, accumulate high levels of defense proteins and exhibit increased resistance to insects [36]. Furthermore, JA has been found to stimulate transcription of genes related to flavonoid biosynthesis and induce anthocyanin accumulation in several plants [37]. Both flavonoid and anthocyanin in plants might act as scavengers to clean redundant free radicals caused by ionizing radiation exposure.

DELLA protein family (also a subfamily of the GRAS family of putative transcriptional regulars) is a group of nuclear growth-restraining proteins that mediate the effect of the phytohormone gibberellin (GA) on growth. GAs are plant hormones that regulate growth and influence various developmental processes, including seed germination, elongation growth and flowering time [38]. The DELLA proteins can restrain growth. GA can promote the degradation of DELLA protein resulting the removal of growth restraint [39]. DELLA inhibits GA action. However, when GA translocates into the nucleus and binds to GID1 receptor, this complex can interact with DELLA resulting the degradation of DELLA [40]. Thus, DELLA proteins appear to play a role in maintaining GA homeostasis. In addition, Dwarf8 (a DELLA protein coding gene) may have a role in flowering time [41] and climate adaptation [42]. Our study showed that the seeds irradiated with 5 kGy grew shorter. It is possible that ionizing radiation disrupted the balance between GA and DELLA, and subsequently led to growth restriction [43].

ζ-carotene desaturase (ZDS) is associated with biosynthesis of carotenoids in plants, and carotenoids are lipid-soluble natural pigments, which may be part of functional protein complexes, such as cytochrome *b6/f* complexes [44]. Carotenoids are active in photosystem assembly and light-harvesting in addition to having photoprotective functions during photosynthesis. They also protect cells from excessive light incidence through thermal dissipation and supply substrates for the biosynthesis of the plant growth regulator abscisic acid, which plays an important role in adaptive responses to environmental stress [45,46].

Increased carotenoid contents can improve the tolerance to stress, such as high light or UV irradiation by scavenging ROS [47]. Carotenoids can be synthesized from the 2-C-methyl-D-erythritol 4-phosphate (MEP) pathway. Two enzymes, ZDS and carotenoid isomerase, are utilized to convert ζ-carotene into lycopene. Next, α-and β-carotene are generated from lycopene and subsequently transformed into lutein and zeaxanthin [48,49]. The transcript levels of ZDS from Gentiana lutea were found to be low in young buds and high in fully open flowers [50]. Similarly, the transcript levels of wheat ZDS were also high in flower petals and were nearly absent in seeds [51].

Since carotenoids can provide protection against oxidative stresses via scavenging ROS, up-regulating ZDS can enhance the resistance to oxidative stresses [47]. Indeed, Li and co-workers found that MDA contents from samples overexpressing ZDS were significantly decreased under salt stress [52]. According to Li’s study, the overexpression of ZDS can significantly increase the amounts of β-carotene and lutein in addition to enhancing salt tolerance [52]. As expected, the protein expression level of ZDS was up-regulated along with down-regulated MDA contents in this study. It has been known that carotenoids can promote human health and protect from aging-related diseases. Carotenoids, having both antioxidant and pro-oxidant properties can manage ROSs and other factors to regulate the apoptosis of cancer cells or the survival of normal cells [53]. Thus, increasing the levels of carotenoids and ZDS in edible plants may both be beneficial in human health and plant survival against harsh conditions.

The yields of crops can be reduced due to abiotic stresses. These environmental stimuli are known be sensed through chloroplasts and mitochondria [54]. DEAD-box RNA helicases (RHs) are involved in essential functions in RNA metabolism where mitochondrial regulation of RNA metabolism by plant nucleus-encoded RNA-binding proteins (RBPs) plays crucial adapting roles against abiotic stresses [55]. DEAD-box ATP-dependent RNA helicase 9 (mitochondrial, with the gene name of RH9) is one of the RBPs. It has been shown that the AtRH9 mRNA level can be enhanced in response to various biotic stresses [56]. In addition, the expression level of AtRH9 was up-regulated in response to cold stress and down-regulated under dehydration and salt conditions [57]. Interestingly, the up-regulated expression of AtRH9 may result the retarded seed germination [57]. This is in concord with our finding, in which irradiated seeds grew slower.

Floral homeotic protein AGAMOUS is a protein functioning as a transcription factor with a molecular weight of 28.7 kDa. AGAMOUS is required during flower development of stamens and carpels [58]. At the early stage of plant development, the expression level of AG needs to be low otherwise early flowering will happen. Functional speaking, AG can be repressed to control the flower patterning and flowering while AG’s activation is involved in flower patterning and floral meristem determinacy [59]. The up-regulated expression of AGAMOUS in irradiated samples may indicate a plant survival mechanism, its ability to produce the offspring. Indeed, plants may adjust the flowering schedule in order to response to irregular environmental conditions [60].

It has been proposed that systemic defense responses can be activated by the peptide systemin and the oxylipin-derived phytohormone jasmonic acid (JA). An interaction of systemin with its receptor (probably Systemin receptor 160, as mentioned above) in the plasma membrane activates JA biosynthesis. After JA is yielded, JAZ (jasmonate ZIM-domain proteins) family repressors will be degraded. This relieves the transcription factors to signal the expression of defense-related genes [61]. Interestingly, in Jibran’s study, plants defective in AGAMOUS expressed much lower JA level compared with that of the wild-type. They suggested that AGAMOUS can induce JA biosynthesis [62]. This indicates that the up-regulated expression of AGAMOUS may help to defend against the stress caused from the irradiation.

Since plants cannot move on their own, it is impossible to escape from various ubiquitous abiotic stresses and adverse environmental conditions (such as salinity, flooding, drought, heat, toxic or heavy metals and irradiation). In the previous studies, it has been indicated that the gamma-ray irradiation can improve and promote the abiotic stress tolerance in plants in addition to be used to screen disease-resistant crop varieties and sterilization [63]. Furthermore, it has been pointed out that the gamma-ray irradiation can regulate the biosynthesis of many secondary key metabolites and osmolytes, and regulate various metabolic activities to generate tolerance to against the environmental stresses [64].

Gamma-ray irradiation mainly uses the element of Co-60 or an electron accelerator to produce and accelerate the irradiation and electron radiation, respectively. It can also be applied after packaging, resulting free of secondary contamination. The gamma-ray irradiation can either damage the DNAs or inhibit the molecular replication. In addition, water molecules within organisms can be ionized or excited to produce harmful free radicals, causing destructive effects. The pathways of ROS metabolism in plant seeds by gamma-ray irradiation are shown in Figure 6. ROS, H_2_O_2_, OH· can be produced by ionizing radiation. These ROS molecules not only attack the phospholipids of cell membranes, but also destroy proteins, nucleic acids and organelles, resulting in cell death or apoptosis. Direct action is mediated by interaction of a secondary electron with the DNA, resulting from absorption of gamma-ray irradiation. Indirect action is mediated by interaction of a secondary electron with water molecules to produce ROS, and then induces DNA damages and lipid peroxidation. Although the gamma-ray irradiation can produce free radicals, plants may still have the ability to self-heal against radiation damage. The interactions of anti-oxidative stress proteins play roles in plant growth, flowering and defense response. The proteomic approach was utilized as a discovery tool to analyze the plant resistance mechanisms and immune response. Those proteins act as transcriptional activators to regulate seed germination, root growth, flowering and plant development.

In agriculture, gamma-ray irradiation can be used to sterilize, kill parasitic pests and inhibit plant growth. After the treatment of gamma-ray irradiation, the nutrients in food may be altered because proteins may be susceptible to free radical energy transfer where proteins are cleaved into peptides or amino acids. Some vitamins are susceptible to the exposure of gamma-ray irradiation. In general, vitamin A, C, E, B-12, thiamine and quinones are quite sensitive to radiation. On the other hand, niacin, pyridoxine, riboflavin, vitamin D, pantothenic acid and biotin are insensitive. In this study, the content of Vitamin P, a flavonoid, was increased.

According to the WHO Technical Report Series 659 and 890 reports, the FAO/IAEA /WHO Expert Committee concluded that foods irradiated by gamma ray to achieve the intended technological objective is both safe to consume and nutritionally acceptable [65]. This conclusion is based on extensive scientific evidence that the preservation process can be used effectively to eliminate spores of proteolytic strains of *Clostridium botulinum* and all spoilage microorganisms. In addition, such treatment does not compromise the nutritional composition of the foods nor cause any toxicological hazard. Therefore, the committee concluded that the irradiation of any food commodity up to an overall average dose of 10 kGy presents no toxicological hazard, no special nutritional or microbiological concern. Therefore, toxicological testing of foods treated in this method is no longer required.

## 4. Conclusions

In order to grow and reproduce, plants are under stress and must strive against a number of detrimental factors. For the direct effect of gamma-ray irradiation, cells lose their physiological metabolism and then decease. The indirect effect, due to the high-energy radiation after cleavage, the water molecules in the cells may be converted into H·, OH·, H_2_O·, H_2_O_2_, or other free radicals. These high-energy free radicals may react with cellular DNA, enzymes, and other proteins causing denaturation and cell death. However, gamma-ray irradiation can also induce the production of anti-oxidants in plants, such as increased flavonoid levels and DPPH radical scavenging activities. In addition, several proteins having anti-oxidant activities were identified. Our findings suggest that these proteins may help seeds strive against the stress of gamma-ray irradiation.

## Figures and Tables

**Figure 1 antioxidants-11-02498-f001:**
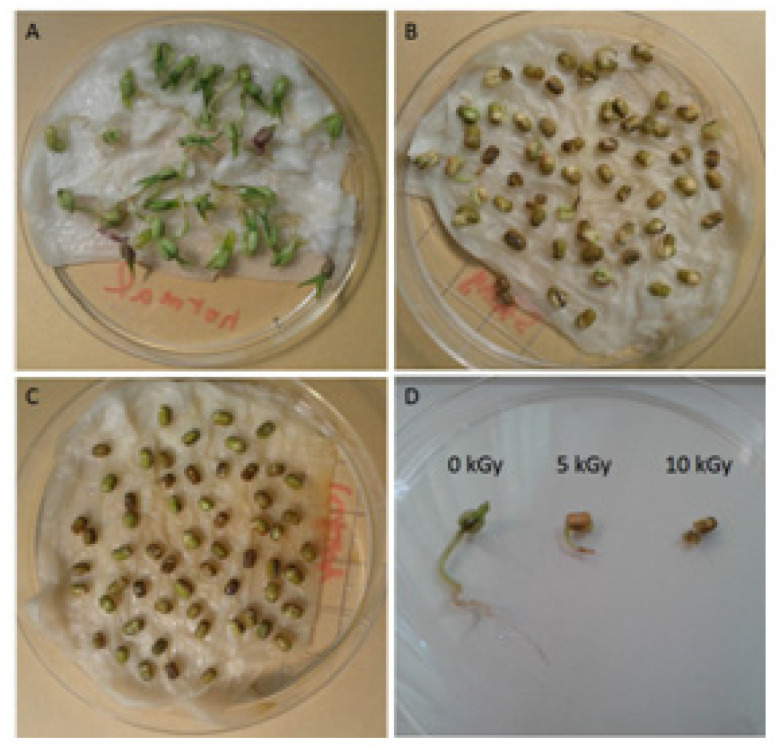
The phenotypic appearance of non-irradiated (0 kGy) and irradiated (5–10 kGy) mung beans. Seeds grown for four days without irradiation (**A**), with 5 kGy irradiation (**B**), or with 10 kGy irradiation (**C**) are shown in groups. For comparison, one seed from each condition is shown in parallel (**D**).

**Figure 2 antioxidants-11-02498-f002:**
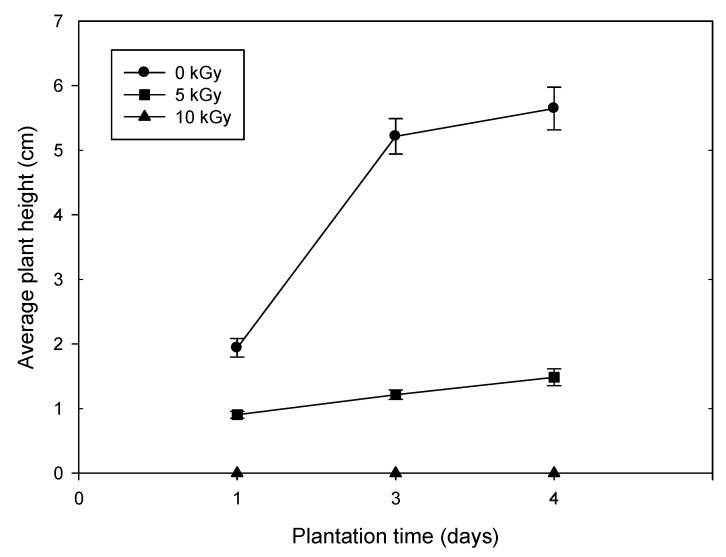
The changes in plant height under various doses of gamma-ray irradiation (mean ± standard deviation, N = 20/group).

**Figure 3 antioxidants-11-02498-f003:**
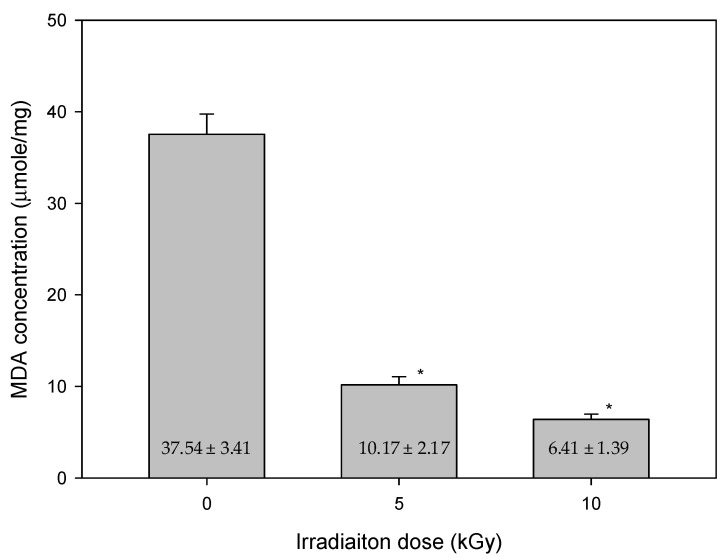
The quantities of MDA, an index for oxidation stress after various gamma-ray irradiation doses of mung beans. MDA content of non-irradiated (0 kGy) and irradiated (5 or 10 kGy) mung beans. (mean ± standard deviation, * *p* < 0.001; N = 20/group).

**Figure 4 antioxidants-11-02498-f004:**
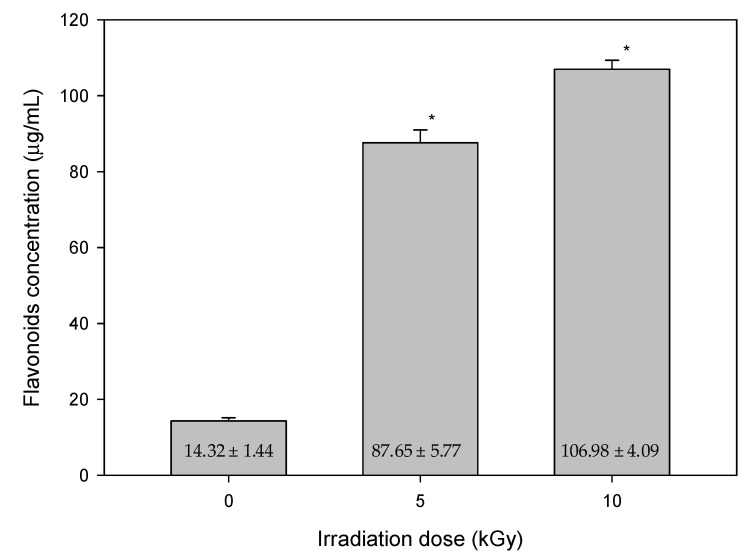
The measurements of total flavonoids of non-irradiated (0 kGy) and irradiated (5 or 10 kGy) mung beans. (mean ± standard deviation, * *p* < 0.001; N = 20/group).

**Figure 5 antioxidants-11-02498-f005:**
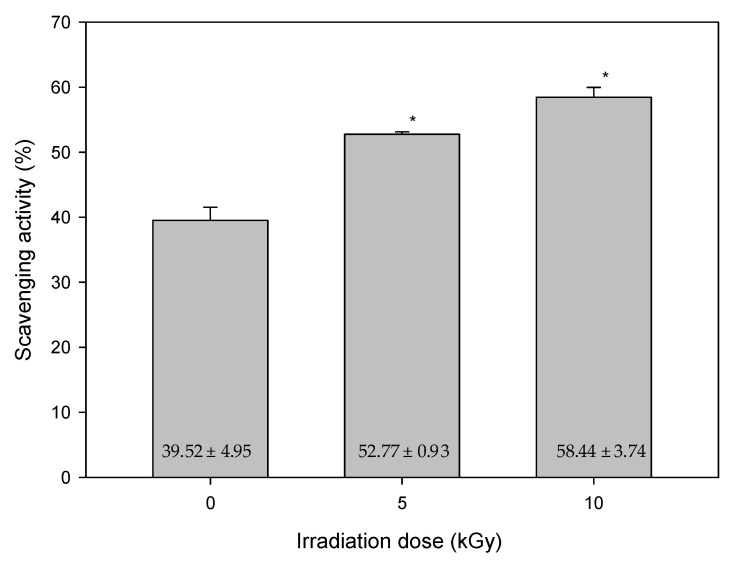
DPPH radical scavenging activities of non-irradiated (0 kGy) and irradiated (5 or 10 kGy) mung beans. (mean ± standard deviation, * *p* < 0.001; N = 20/group).

**Figure 6 antioxidants-11-02498-f006:**
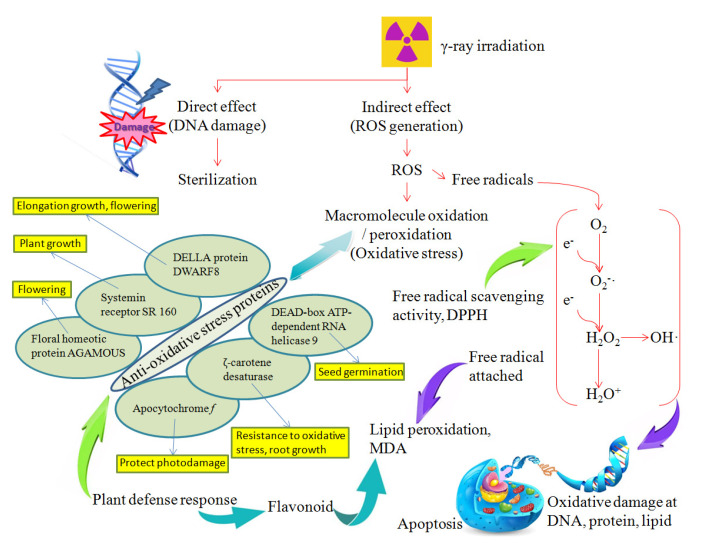
The effect of gamma-ray irradiation on the biological system of plant. Radiation can cause direct and indirect effects to DNA and proteins. For indirect effects, ROS are generated leading to increased oxidative stress and free radicals. In response to these radiation effects, flavonoid and anti-oxidative stress proteins are induced via this proposed mechanism.

**Table 1 antioxidants-11-02498-t001:** After 5 kGy gamma-ray irradiation, the peptide sequence tags, subcellular localization and protein functional classification of six up-regulated proteins with higher confidence levels (at least two unique peptide sequences matched) were identified.

Swiss-Port/TrEMBL Accession Number	Protein Name	MW (Da)	Score	Match Queries	Sequence Coverage (%)	Match Peptide *	Subcellular Location	Function
Q1ACM5	Apocytochrome f	35,148	31	16	8	R.IVCANCHLAKK.T R.GQIYLDGSKSNNTIYSSSAEGQVVK.I K.ISLKLLK.T R.IVCANCHLAKK.S K.KNIVVVGPIPGK.T R.IQGLIAFFISVIIAQTFLVLKKK.Q K.GGLNVGAVLILPEGFQIAPADRIPEEMKSK.I R.IVCANCHLAKK.S K.GGLNVGAVLILPEGFELAPSDRISPEIKQK.I -.MQTAITKK.W -.MQTAITK.K R.IVCANCHLAKK.T K.SNNTVYTASATGK. I R.IVCANCHLAKK.A K.IPYDTQIKQVLSNGKK.G K.SNNNVFSASTAGTISQITR.Q	Plastid	Component of the cytochrome b6-f complex, which mediates electron transfer between photosystem II (PSII) and photosystem I (PSI), cyclic electron flow around PSI, and state transitions
Q9ST48	DELLA protein DWARF8	65,988	35	6	7	R.SSDMADVAQK.L R.EYQDAGGSGGDMGSSK.D R.MRTGGGSTSSSSSSSSSMDGGR.T R.MRTGGGSTSSSSSSSSSMDGGR.T R.MRTGGGSTSSSSSSSSSMDGGR.T R.MRTGGGSTSSSSSSSSSMDGGRTR.S	Nucleus	Probable transcriptional regulator that acts as a repressor of the gibberellin (GA) signaling pathway. Probably acts by participating in large multiprotein complexes that repress transcription of GA-inducible genes. Upon GA application, it is degraded by the proteasome, allowing the GA signaling pathway.
Q9LUW6	DEAD-box ATP-dependent RNA helicase 9	63,609	25	4	11	R.SSFGGFGSNDGK.R K.SLPSNSSPFGVKVR.D R.SGGGGYGSYGSSSGR.S R.SGGGSYGGYGGSSGRSGGGGGSYGGSGGSSSR.Y	mitochondrion	RH9 was participated in RNA metabolism, which associated with the cellular function of response to abiotic stress.
Q40168	Floral homeotic protein AGAMOUS	28,706	20	5	31	K.NLLKKIYK.L R.IEKGISKIR.S K.LRAQIGNLMNQNR.N K.ACSDSSNTGSVSEANAQYYQQEASK.L R.AQHQHQQMNLMPGSSSNYHELVPPPQQFDTR.N	Nucleus	Probable transcription factor involved in the control of organ identity during the early development of flowers. Is required for normal development of stamens and carpels in the wild-type flower. Plays a role in maintaining the determinacy of the floral meristem. Acts as C class cadastral protein by repressing the A class floral homeotic genes such as APETALA1. Forms a heterodimer via the K-box domain with either SEPALATTA1/AGL2, SEPALATTA2/AGL4, SEPALLATA3/AGL9 or AGL6 that could be involved in genes regulation during floral meristem development.
Q8L899	Systemin receptor SR160	131,880	31	4	4	K.AQLKDGSVVAIKK.L K.QSGNIAVALLTGKR.Y K.LPVDTLLKLSNIK.T K.EDASIEIELLQHLK.V	Cell membrane; Single-pass type I membrane protein.	Receptor with a serine/threonine-protein kinase activity. Involved in the perception of systemin, a peptide hormone responsible for the systemic activation of defense genes in leaves of wounded plants. May also regulate, in response to brassinosteroid binding, a signaling cascade involved in plant development
O49901	ζ-carotene desaturase, chloroplastic/chromoplastic	63,542	30	5	7	R.LPMGAPLHGIR.A K.TPVKNFFLAGSYTK.Q -.MASSTCLIHSSSFGVGGKK.V K.ADVYIAACDVPGIK.R K.SANGETYVTGLAMSK.A	Plastid, chloroplast, chromoplast	Catalyzes the conversion of zeta-carotene to lycopene via the intermediary of neurosporene. It carries out two consecutive desaturations (introduction of double bonds) at positions C-7 and C-7’

* Underline: Protein post-translational modification (PTM), C: Carboxymethyl; NQ: Deamidated; M: Oxidation; STY: Phospho.

## Data Availability

The data that support the findings of this study are contained within the article. More information is available on request from the corresponding author.

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
