# Peer review of "Utilizing Proteomic Approach to Analyze Potential Antioxidant Proteins in Plant against Irradiation"

_antioxidants, 2022, doi:10.3390/antiox11122498_

Round 1

Reviewer 1 Report

The proteomic approach should produce more results with more accuracy. In a broader look, there is not much that we can concluded through this approach as we are only considering the protein expression.

Author Response

The proteomic approach should produce more results with more accuracy. In a broader look, there is not much that we can concluded through this approach as we are only considering the protein expression.

Reply: The authors appreciated that the Reviewer pointed out this comment.

In the protein identifications of the proteomic study, it is indeed true that only the differential expression of proteins was considered, as was pointed out by the reviewer. However, we set a stricter identification threshold for the protein identifications. These proteins passed the high threshold meaning that they are reliable. Since there is no suitable antibody for plant proteins to perform Western blot analysis, the database search was used to analyze the correlation of these identified proteins and confirm their role in gamma-ray irradiation. For the threshold of protein identification, we have added a paragraph in section 3.3 (Page 9), and the protein functions are shown in Figure 6 in the page15-16.

Reviewer 2 Report

The manuscript entitled "Utilizing proteomic approach to analyze potential antioxidant proteins in plant against irradiation " intended for publication in Antioxidants is relevant to the Journal, and Special Issue: Antioxidants in Food and Cosmetics; manuscript is generally an interesting paper, however in the present form is not ready to publish and needs improvements.

The paper is generally straightforward, but some parts of the manuscript need much more attention. Authors should improve Keywords to avoid word repetition from the title, and should explain all abbreviations used in the text of manuscript. I think that Introduction could be longer and more in detail and the Authors should more specify/improve the purpose of the study. In addition, Authors must improve Material and methods section, provide more on plant material and growth condition (e.g. explain: “room temperature with normal photoperiods”). The presentation and interpretation of results is not always clear. I think, that Authors should improve figures presentation – e.g. Figs. 1-2 and Fig. 3 (check units). The Conclusions could be longer and modified, to really focused on results. Also the choice of references could be reconsidered for factual representation of the current progress in the field, there is much more papers on the subject. Authors should check more carefully the Reference list, and improve it.  In addition, there are many small mistakes in the text of manuscript, including Reference list, that need to be corrected by Authors (e.g. lines: 33, 70, 155, 185, 188, 216, 224, 230, 238, 259, 292, 316, 359, 413, 423, 471, 473, 480, 509, 526, 534, 538).

Author Response

The manuscript entitled "Utilizing proteomic approach to analyze potential antioxidant proteins in plant against irradiation " intended for publication in Antioxidants is relevant to the Journal, and Special Issue: Antioxidants in Food and Cosmetics; manuscript is generally an interesting paper, however in the present form is not ready to publish and needs improvements.
The paper is generally straightforward, but some parts of the manuscript need much more attention. Authors should improve Keywords to avoid word repetition from the title, and should explain all abbreviations used in the text of manuscript. I think that Introduction could be longer and more in detail and the Authors should more specify/improve the purpose of the study. In addition, Authors must improve Material and methods section, provide more on plant material and growth condition (e.g. explain: “room temperature with normal photoperiods”). The presentation and interpretation of results is not always clear. I think, that Authors should improve figures presentation – e.g. Figs. 1-2 and Fig. 3 (check units). The Conclusions could be longer and modified, to really focused on results. Also the choice of references could be reconsidered for factual representation of the current progress in the field, there is much more papers on the subject. Authors should check more carefully the Reference list, and improve it.  In addition, there are many small mistakes in the text of manuscript, including Reference list, that need to be corrected by Authors (e.g. lines: 33, 70, 155, 185, 188, 216, 224, 230, 238, 259, 292, 316, 359, 413, 423, 471, 473, 480, 509, 526, 534, 538).

Reply: The authors appreciated that the Reviewer pointed out this comment. We modified the keywords and explained all abbreviations used in the text of manuscript. The Introduction and Material and methods were improved. The plant material and growth condition were also provided (Section 2.1, Page 3). Figures 1 to 3 were modified. The units for figures were checked. According to the report "Wholesomeness of irradiated food: report of a Joint FAO/IAEA/WHO Expert Committee [‎meeting held in Geneva from 27 October to 3 November 1980]‎, the committee determined that the irradiation of any food commodity up to an overall average dose of 10 kGy presents no toxicological hazard and, moreover, that the toxicological testing of foods so treated was no longer required. The report is recognized as a landmark in that it established a maximum absorbed dose, e.g. 10 kGy, for the wholesomeness of irradiated food. Discussion and Conclusions sections were modified; we added new Figure 6 and some paragraphs to explain our finding (Page 15). Also, several new references were added in the revised manuscript. Besides, those small mistakes were corrected by Authors. Thank you for reminding.

Round 2

Reviewer 1 Report

The use of other proteomic tools should be addressed in order to obtain more identified proteins and higher scores.

Author Response

The use of other proteomic tools should be addressed in order to obtain more identified proteins and higher scores.

Reply: The authors appreciated that the Reviewer pointed out this comment. In the past, the aim of proteomic analysis was to analyze a large number of proteins in a short period of time. However, in this experiment, we do not seek for discovering a lot of proteins. Instead, we are working on finding the key differential expressed proteins. Thus, a higher threshold was set for this study. In this study, we adapted two mass spectrometry analysis software: Bioworks 3.1 (Thermo Finnigan) and Mascot 2.3 (Matrix Science). Mascot is the most commonly employed search engine software that utilizes mass spectrometry data to identify proteins from peptide sequence databases. The latest version of protein database (UniProtKB/Swiss-Prot databases) can be downloaded and updated from NCBI website. On the other hand, Bioworks 3.1 is an older version of the built-in database in our laboratory. Since it does not contain the plant database, the Bioworks workstation was used to assess the quality of the mass spectra in this experiment. Thus, these proteins were identified by MASCOT software after being examined as high quality mass spectra. As a result, there are not a lot of proteins were identified in this experiment, but all of them were highly reliable. To make it unambiguous, we have revised the description again in the revised manuscript. Due to the lack of research funding, there are only two sets of mass spectrometry analysis software available in our laboratory. We may purchase an updated version of the analysis software in the future to improve our proteomic data analysis if more funding is granted.